# Genome-scale metabolic models for natural and long-term drug-induced viral control in HIV infection

Anoop T Ambikan[1,*], Sara Svensson-Akusjärvi[1,*], Shuba Krishnan[1], Maike Sperk[1], Piotr Nowak[1,2], Jan Vesterbacka[2], Anders Sönnerborg[1,2], Rui Benfeitas[3], Ujjwal Neogi[1,4]

Genome-scale metabolic models (GSMMs) can provide novel insights into metabolic reprogramming during disease progression and therapeutic interventions. We developed a context-specific system-level GSMM of people living with HIV (PLWH) using global RNA sequencing data from PBMCs with suppressive viremia either by natural (elite controllers, PLWH_EC) or drug-induced (PLWH_ART) control. This GSMM was compared with HIV-negative controls (HC) to provide a comprehensive systems-level metabo-transcriptomic characterization. Transcriptomic analysis identified up-regulation of oxidative phosphorylation as a characteristic of PLWH_ART, differentiating them from PLWH_EC with dysregulated complexes I, III, and IV. The flux balance analysis identified altered flux in several intermediates of glycolysis including pyruvate, α-ketoglutarate, and glutamate, among others, in PLWH_ART. The in vitro pharmacological inhibition of OXPHOS complexes in a latent lymphocytic cell model (J-Lat 10.6) suggested a role for complex IV in latency reversal and immunosenescence. Furthermore, inhibition of complexes I/III/IV induced apoptosis, collectively indicating their contribution to reservoir dynamics.

## Introduction

During HIV-1 infection, cellular metabolic activity, combined with glycolytic enzymes, regulates susceptibility to HIV-1 at the cellular level (Clerc et al, 2019; Valle-Casuso et al, 2019). Elevated oxidative phosphorylation (OXPHOS) and glycolysis thus favor infection in CD4+ T cells (Clerc et al, 2019; Valle-Casuso et al, 2019). CD4+ T cells up-regulate glycolysis to meet the energy-demanding turnover for HIV-1 virion production, resulting in their eventual death (Hegedus et al, 2014; Palmer et al, 2014). After initiation of combination antiretroviral therapy (cART), virus-induced short-term metabolic changes do not restore the transient metabolic modulation caused by the infection. Altered amino acid (AA) metabolism has been reported in untreated people living with HIV-1 (PLWH) as well as within the first 2 yr after initiation of cART compared with the HIV-negative controls (Cassol et al, 2013; Peltenburg et al, 2018). In our recent extensive multi-omics system biology studies on cohorts from India (Babu et al, 2019; Gelpi et al, 2021), Cameroon (Gelpi et al, 2021), and Denmark (Gelpi et al, 2021; Villumsen et al, 2022), we mapped the in-depth metabolomic dysregulation associated with long-term treatment in PLWH. Our group (Babu et al, 2019; Gelpi et al, 2021; Villumsen et al, 2022), and others (Mukerji et al, 2016; Rosado-Sánchez et al, 2019; Valle-Casuso et al, 2019; Meeder et al, 2021; Shytaj et al, 2021), have highlighted how the coordinated modulation of central carbon metabolism, AA metabolism, glutaminolysis, and fatty acid biosynthesis can potentiate accentuated immune aging and cognitive decline in a subset of PLWH on therapy who have dysregulated metabolic profile.

Elite controllers (EC) are a unique group of PLWH that naturally control viral replication and maintain a low viral reservoir. Our recent study indicated that EC had a distinct lipid profile, reduced inflammation, and increased antioxidant defense which may contribute to the EC status (Akusjärvi et al, 2022). Moreover, the integrative proteomic and transcriptomic analysis suggested that the EC group had a unique metabolic uptake and flux profile through hypoxia-inducible factor signaling and glycolysis, which could contribute to the natural control of HIV-1 infection (Akusjärvi et al, 2022). A study also showed how suboptimal inhibition of glycolysis in CD4+ T cells decreased the latently infected reservoir (Valle-Casuso et al, 2019). However, EC is heterogeneous, and one mechanism of elite control does not exist (Zhang et al, 2018; Akusjärvi et al, 2022). Instead, PLWH on long-term successful therapy with prolonged suppressive viremia are more homogenous in their immune profile (Zhang et al, 2017). A deep understanding of the immune profile of these groups of HIV-1–infected individuals could help to develop strategies for analytical treatment interventions (ATI) to achieve a clinically relevant ART-free HIV cure or remission (Julg et al, 2019).

[1]Division of Clinical Microbiology, Department of Laboratory Medicine, Karolinska Institutet, ANA Futura, Campus Flemingsberg, Stockholm, Sweden   [2]Department of Medicine, Huddinge (MedH), Karolinska Institutet, ANA Futura, Campus Flemingsberg, Stockholm, Sweden   [3]National Bioinformatics Infrastructure Sweden (NBIS), Science for Life Laboratory, Department of Biochemistry and Biophysics, Stockholm University, Stockholm, Sweden   [4]Manipal Institute of Virology (MIV), Manipal Academy of Higher Education, Manipal, Karnataka, India

Correspondence: ujjwal.neogi@ki.se
*Anoop T Ambikan and Sara Svensson-Akusjärvi contributed equally to this work.

Genome-scale metabolic models (GSMMs) can provide novel insights toward understanding host–pathogen interactions and metabolic reprogramming during acute and chronic infections. When applied to PBMCs, GSMM can contribute to unraveling the mechanistic processes at the systems level (Ambikan et al, 2022). By combining contextualized PBMC-specific biological network analysis, GSMMs, and multi-omics integration, one can attain holistic and dynamic characterizations of complex rearrangements during disease progression or therapeutic interventions (Yang et al, 2021; Zeybel et al, 2021).

In the present study, we sought to understand and infer changes in HIV-1 infection at the system level by comparing successfully treated PLWH on prolonged cART (herein PLWH$_{ART}$) with the HIV-seropositive ECs (herein PLWH$_{EC}$) and HIV-negative controls (herein HC). Contextualized PBMC-specific GSMMs and biological networks were thus developed for PLWH$_{ART}$ and PLWH$_{EC}$ to identify the metabolic alterations during prolonged therapy. We further modulated the key pathways pharmacologically to determine their role in HIV-1 reservoir dynamics and immune senescence profile. By combining the multidimensional omics data, our study is the first to provide a comprehensive mapping of the immunometabolic dysregulations using GSMM in PLWH$_{ART}$ with successful long-term treatment. Furthermore, our comparative analysis with PLWH$_{EC}$ offers mechanistic insights into natural immune control.

# Results

### Clinical characteristics

The study population included three PLWH cohorts, where two groups had suppressed viremia (PLWH$_{ART}$ and PLWH$_{EC}$, n = 19 each), and one untreated group was viremic (herein PLWH$_{VP}$, n = 19). In addition, we included 19 HC. Given that extensive transient metabolic changes occur in the PLWH$_{VP}$ due to the acute viremic phase, we used this group to develop the cART-specific model only (see the Materials and Methods section). The clinical characteristics are given in Table S1. The median (IQR) duration of diagnosed HIV-1 seropositivity infection for PLWH$_{EC}$ was 9 (5–14) yr, and none had received treatment. In PLWH$_{ART}$, the median duration of suppressive therapy was 13 (7–17) yr with no viral blips except for two individuals. At the time of sample collection, both PLWH$_{ART}$ and PLWH$_{EC}$ had undetectable plasma viral load (<20 copies/ml) and CD4$^+$ T-cell counts (>500 cells/$\mu$l) indicative of significant immune reconstitution.

### System-level PBMC-based gene expression identifies dysregulation of OXPHOS in PLWH$_{ART}$

To identify the system-level host response during HIV-1 infection, we performed transcriptomic profiling of total RNA isolated from PBMCs. The differential gene expression analysis was performed between all pair-wise comparisons among the four groups (adjusted $P < 0.05$, Supplemental Data 1). No genes were found to be dysregulated between PLWH$_{EC}$ and HC, whereas 949 genes were differentially expressed between PLWH$_{ART}$ and HC (adjusted $P < 0.05$). To identify whether the changes in gene expression between

the groups were due to the altered cell-type proportions, we performed digital cell quantification (DCQ), estimating cell-type proportions in each group (Racle & Gfeller, 2020). We characterized 18 immune cell populations (Figs 1A and S1). As expected, the proportion of several cell types was significantly different in PLWH$_{VP}$ compared with the other groups. No significant difference in the proportion of cell types was observed between PLWH$_{EC}$ and HC and the only difference in regulatory T cells (Tregs) was observed between PLWH$_{EC}$ and PLWH$_{ART}$ ($P < 0.05$). Based on the differentially expressed genes, we identified 1,037 specifically dysregulated genes in PLWH$_{ART}$ (see the Materials and Methods section) with an explicitly differential regulation in PLWH$_{ART}$ (Supplemental Data 2). Sample clustering using the cART-specific genes separated PLWH$_{ART}$ samples from the other groups (Fig 1B). One PLWH$_{ART}$ sample (marked by an arrow) (Fig 1B) was identified as belonging to a patient who had been classified in the past as an EC but started treatment 23 yr after HIV-1 diagnosis (date of diagnosis: 05 January, 1989, treatment initiation 11 December, 2012) due to two successive viral loads were above the detection limit (240 and 185 copies/ml, respectively). The patient maintained viral load below detection level following treatment. Gene set enrichment analysis (GSEA) using MSigDB hallmark gene sets on the ART-specific genes highlighted that the primary mechanisms related to the long-term treatment were OXPHOS (adjusted $P < 0.05$) and reactive oxygen species (ROS) pathways (adjusted $P < 0.1$), as the top significantly regulated gene sets (Fig 1C).

### Larger viral reservoir and up-regulated OXPHOS differentiate PLWH$_{ART}$ from PLWH$_{EC}$

Next, we performed a comparative analysis between PLWH$_{EC}$ and PLWH$_{ART}$ to identify the immune signature during suppressive viremia that is naturally controlled, or cART induced. First, we performed relative reservoir quantification on total PBMC HIV-1 DNA and observed that PLWH$_{ART}$ had a significantly larger reservoir than PLWH$_{EC}$ (Fig 2A). Furthermore, we performed differential gene expression analysis between PLWH$_{EC}$ and PLWH$_{ART}$ to identify the cART related changes during suppressive viremia. We identified 1,061 significantly dysregulated genes in PLWH$_{ART}$ compared with PLWH$_{EC}$ (adjusted $P < 0.05$). There were 400 genes up-regulated and 661 genes down-regulated in PLWH$_{ART}$ compared with PLWH$_{EC}$ (Fig 2B). The dysregulated genes displayed distinct expression patterns in the two groups and hierarchical clustering, showing apparent clustering of PLWH$_{ART}$ and PLWH$_{EC}$ samples (Fig 2C). No other factors like age, duration of treatment, and gender showed any clustering pattern. The GSEA analysis using the MSigDB hallmark gene set showed that OXPHOS and ROS pathways were significantly enriched with most of the genes up-regulated in PLWH$_{ART}$ (Fig 2D) (false discovery rate [FDR] < 0.2). Pathways with most genes down-regulated in PLWH$_{ART}$ were not statistically significant. Pathways such as mTORC1 signaling and glycolysis also appeared in the analysis, with most of the genes up-regulated in PLWH$_{ART}$ but without passing the significance threshold (FDR > 0.2). OXPHOS was identified as significantly altered in long-term treated patients. Therefore, we looked at OXPHOS in detail to find which complexes were most affected. Among the genes in the five complexes of OXPHOS (I to V), complexes I (34%), III (45%), and IV (45%) were

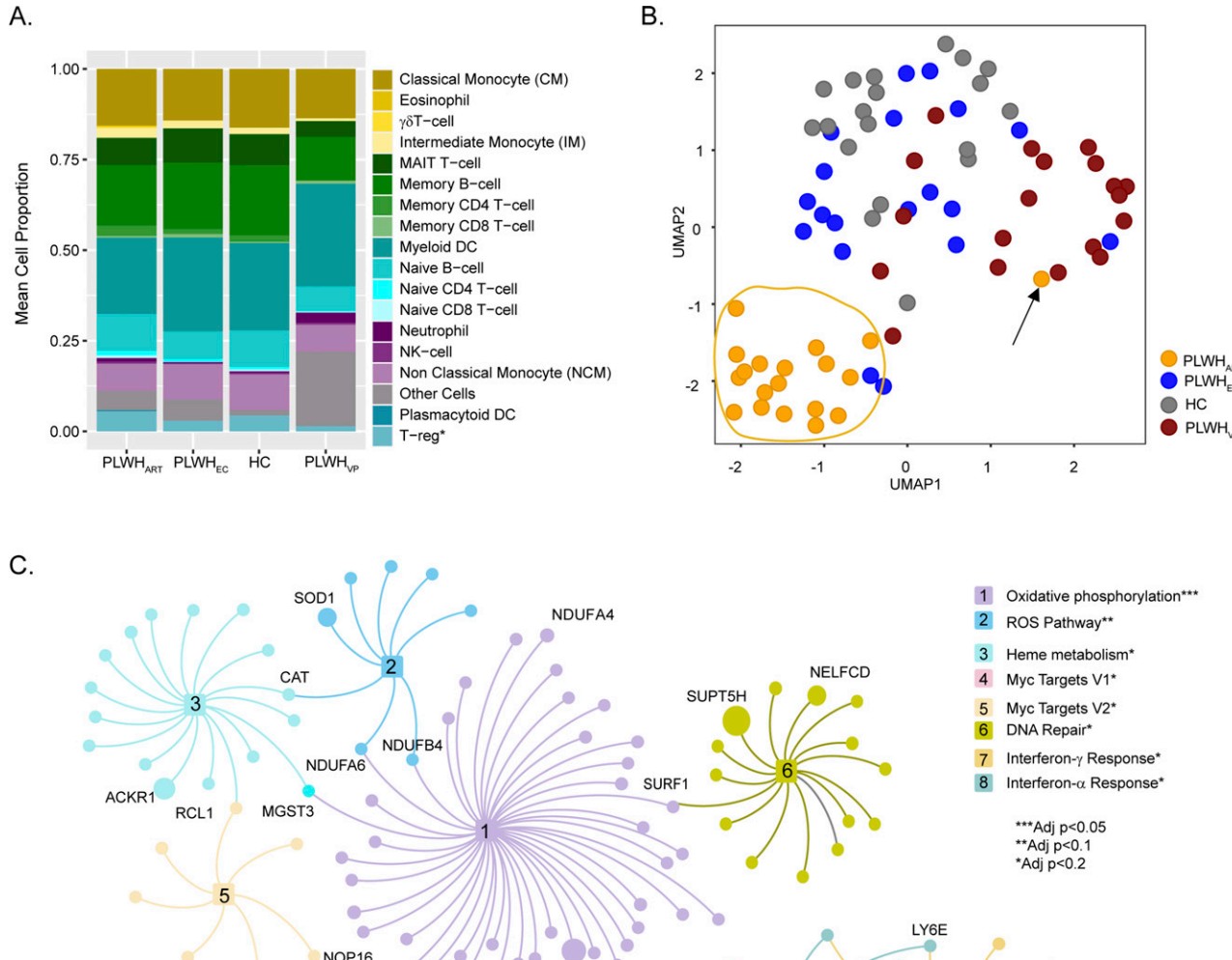

**Figure 1. System-level transcriptomics signature in PLWH_ART.**
**(A)** Digital cell-type quantification using Estimating the Proportions of Immune and Cancer cell (EPIC) methodology. Mean cell proportions estimated from the samples of each of the four cohorts are visualized in the bar graph. **(B)** Visualization of sample distribution using expression of combination antiretroviral therapy–specific genes and dimensionality reduction by UMAP. **(C)** Network visualization of pathways identified as significantly enriched by combination antiretroviral therapy–specific genes. Nodes are genes and edges represent association with pathways. Node size is relative to the mean expression of the genes among the PLWH_ART. Genes overlapping between pathways and high abundance genes are labeled.

primarily affected in PLWH_ART compared with PLWH_EC (Fig 2E). We also checked the overlap between the ART-specific genes (n = 1,037) and the differentially regulated genes between the PLWH_ART and PLWH_EC (n = 1,061). We observed that 557 genes were overlapping between the two sets of genes. The gene list enrichment analysis using MSigDB hallmark gene set identified OXPHOS (adjusted $P <$ 0.001), MYC targets V1 (adjusted $P$ = 0.004), and ROS pathway (adjusted $P$ = 0.04) as significant pathways. Combining all the data, up-regulation of the OXPHOS was the hallmark of PLWH_ART and complexes I, III, and IV were primarily affected.

## Altered flux balance in PLWH on cART is linked to OXPHOS, glycolysis, and TCA cycle

Given that significant metabolic pathway-centered perturbations were found in PLWH_ART, we next performed reporter metabolite analysis to identify metabolites around which most of the transcriptional changes occurred. Five metabolites, namely, superoxide, ubiquinol, ubiquinone, ferrocytochrome C, and ferricytochrome C were significantly up-regulated in PLWH_ART compared with PLWH_EC (adjusted $P <$ 0.2) (Fig 3A). In addition, nicotinamide adenine

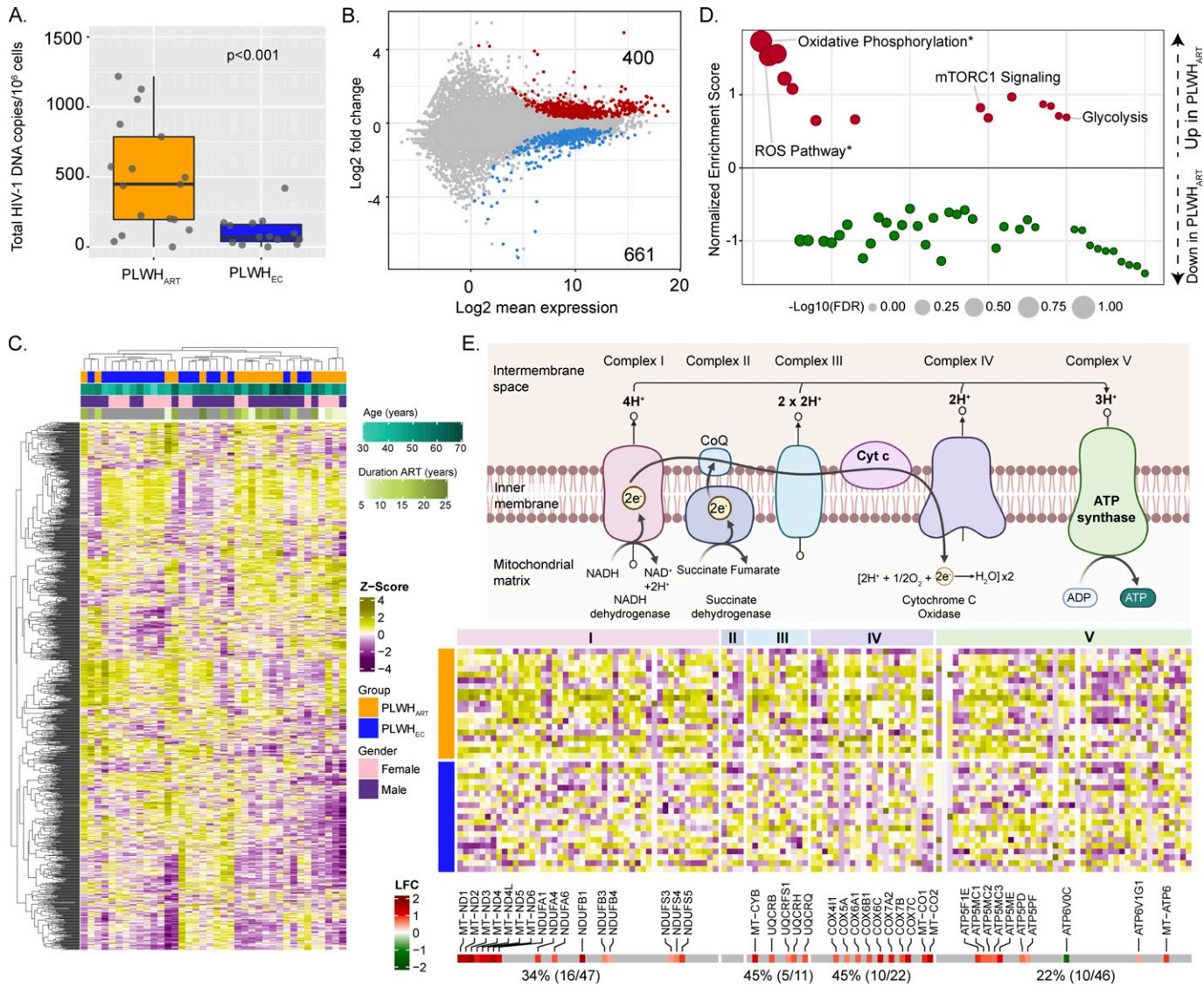

**Figure 2. Comparative analysis of PLWH_ART and PLWH_EC.**
**(A)** Relative reservoir quantification using total HIV-1 DNA in PLWH_ART (n = 17) and PLWH_EC (n = 14). **(B)** MA plot showing differential gene expression results of PLWH_ART versus PLWH_EC. Negative log₂-fold change values represent down-regulation and positive values represent up-regulation in PLWH_ART. Grey-colored dots denote non-significant genes (adjusted $P > 0.05$). **(C)** Heatmap showing the expression pattern of significantly regulated genes between PLWH_ART and PLWH_EC (adjusted $P < 0.05$). Column annotation denotes cohort, age, gender, and duration of combination antiretroviral therapy of the corresponding samples. Row and column clustering was performed using Euclidian distance. **(D)** Gene set enrichment analysis results using MSigDB hallmark gene set between PLWH_ART versus PLWH_EC. A positive enrichment score represents up-regulation and negative score represents down-regulation in PLWH_ART. Statistically significant pathways are labeled and highlighted by asterisk. Bubble size is relative to the adjusted $P$-values of the pathways. *Indicates FDR < 0.2. **(E)** Schematic visualization of the five complexes of OXPHOS pathway. The heatmap shows expression pattern of genes belonging to OXPHOS pathway in PLWH_ART and PLWH_EC. Column annotation denotes OXPHOS pathway complexes and row annotation denotes the cohort. The bottom annotation shows the log₂ fold change values of the genes. Red color represents up-regulation and green color represents down-regulation of the gene in PLWH_ART compared with PLWH_EC.

dinucleotide hydrogen, S-adenosyl methionine, and S-adenosyl-L-homocysteine were predicted to be significantly dysregulated in PLWH_ART (adjusted $P < 0.2$) (Fig 3A). The overall results suggested a significant change in porphyrin, glycine, serine, and threonine metabolism, and a positive regulation in OXPHOS. The reactions involving significant reporter metabolites, catalyzed by genes in complexes I, III, and IV of OXPHOS (Fig S2), had a distinct expression pattern in PLWH_ART compared with PLWH_EC. Next, we performed context (disease state)-specific GSMM and flux balance analysis

(FBA) to calculate the metabolic flux in response to transcriptional changes in the PLWH_ART, PLWH_EC, and HC cohorts (Fig 3B). Context-specific metabolic models for PLWH_ART, PLWH_EC, and HC having 6,179, 6,237, and 6,199 reactions and 1,799, 1,842, and 1,834 genes/transcripts catalyzing them, respectively, were developed (available: github.com/neogilab/LongART). After excluding the reactions with same directional fluxes in all the three cohorts and reactions with insignificant flux ($<10^{-5}$ mmol/h/gDCW), 80 reactions (Supplemental Data 3) were found to be uniquely regulated in PLWH_ART

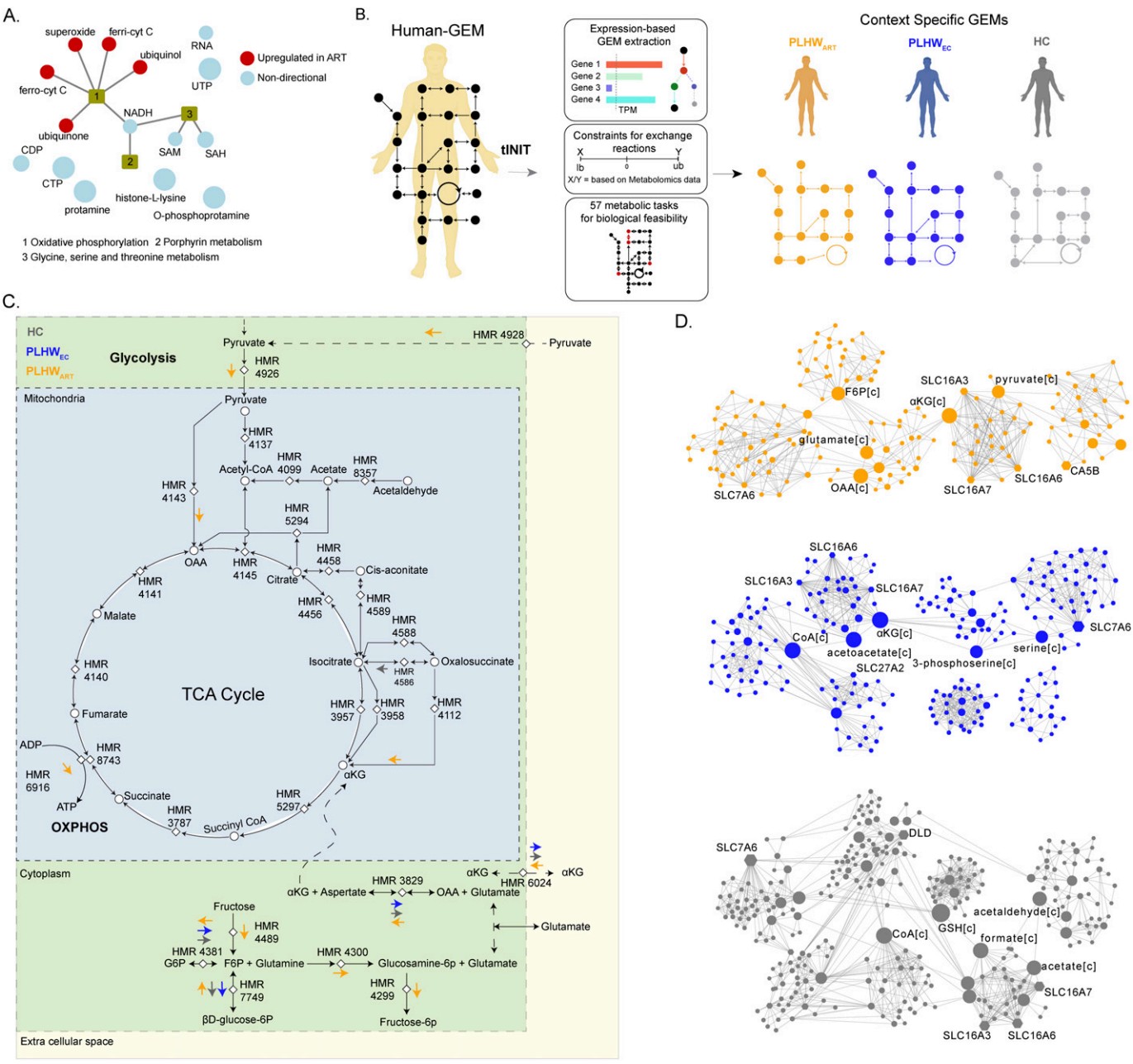

**Figure 3. Context-specific genome-scale metabolic modeling and flux balance analysis.**
**(A)** Network visualization of significant reporter metabolites (adjusted *P* < 0.2) identified in PLWH$_{ART}$ versus PLWH$_{EC}$. Red-colored nodes represent up-regulated reporter metabolites and steel-blue colored nodes represent dysregulated (non-directional) reporter metabolites. **(B)** Workflow diagram of context-specific genome-scale metabolic model reconstruction. **(C)** Reaction diagram showing flux balance analysis results. Reactions show specific flux changes in PLWH$_{ART}$ compared with PLWH$_{EC}$ and HC cohorts highlighted with colored arrows. The direction of the arrow represents the flux change of the corresponding reaction in the cohort. **(D)** Communities identified from the topology analysis of the metabolic network in PLWH$_{ART}$, PLWH$_{EC}$, and HC. Node size is relative to betweenness centrality measurement. The top five ranked genes and metabolites based on betweenness centrality are labeled.

compared with PLWH$_{EC}$ and HC cohorts. These reactions belonged to the AA, nucleotide, carbohydrate, and energy metabolism pathways. There were also 33 significant transport reactions that were transporting metabolites between cell compartments. Of the energy metabolism, pathways surrounding the tricarboxylic acid (TCA) cycle, including glycolysis, glutaminolysis, and OXPHOS, were affected in PLWH$_{ART}$ (Fig 3C). The OXPHOS reaction converting ADP

to ATP (HMR-6916) had a positive flux in PLWH$_{ART}$ whereas no flux was shown in PLWH$_{EC}$ and HC, indicating that higher energy was required in PLWH$_{ART}$. There were also cytoplasmic reactions that appeared to increase the production of α-ketoglutarate (αKG) in PLWH$_{ART}$. Reactions producing fructose-6-phosphate (HMR-7749 and HMR-4489), which further feeds the reaction producing glutamate (HMR-4300), showed a positive flux in PLWH$_{ART}$,

whereas showing a negative or no flux in PLWH$_{EC}$ and HC, respectively, indicative of higher glutamate production and conversion in PLWH$_{ART}$. The reaction converting glutamate and oxaloacetic acid (OAA) to αKG and aspartate (HMR-3829) also showed a flux towards αKG production in PLWH$_{ART}$ and the opposite direction in PLWH$_{EC}$ and HC. Also, the transporter reaction (HMR-6024) transporting αKG from extracellular space to the cytoplasm showed a flux in PLWH$_{ART}$. The reactions mentioned above signify increased accumulation of αKG in the cytoplasm in PLWH$_{ART}$ that can feed the TCA cycle in the mitochondria. The reaction producing OAA from pyruvate (HMR-4143) and the reaction producing αKG from oxalosuccinate (HMR-4112) had positive flux in PLWH$_{ART}$ indicating activation of the TCA cycle. To further understand the metabolic rearrangements, we performed a topological analysis on metabolic networks generated for PLWH$_{ART}$, PLWH$_{EC}$, and HC cohorts. The metabolic networks were generated by drawing edges between reactants, products, and associated genes of the reactions found to exhibit significant (>10$^{-5}$ mmol/h/gDCW) and diverging flux among the three cohorts. Communities were identified and betweenness centrality of the nodes was calculated to rank the genes and metabolites for their influence in the network. The top five metabolites and genes in PLWH$_{ART}$, PLWH$_{EC}$, and HC based on node centrality measurements are shown in Fig 3D. The metabolites fructose-6-phosphate, OAA, glutamate, and pyruvate uniquely play a central role in PLWH$_{ART}$ indicative of a role of TCA cycle and glycolysis in differentiating PLWH$_{ART}$ from PLWH$_{EC}$ and HC. Transporter genes of the SLC16 gene family (monocarboxylate transporters) SLC16A3, SLC16A6, and SLC16A7 were central in all three groups further suggesting a role for pyruvate and lactate transport. Combining these results, it can be concluded that reactions surrounding the TCA cycle including glutaminolysis, OXPHOS, and glycolysis differentiate PLWH$_{ART}$ from PLWH$_{EC}$ and HC.

### Long-term cART disrupts redox homeostasis in the lymphocytic cell population

The earlier used antiretrovirals frequently induced severe adverse effects that were linked to the occurrence of oxidative stress and mitochondrial damage (Smith et al, 2017). As we observed an up-regulation of superoxide, ubiquinol, ubiquinone, ferricytochrome C, and ferrocytochrome C in PLWH$_{ART}$ we evaluated total cellular ROS levels in different PBMCs subpopulations from PLWH$_{ART}$ (n = 16), PLWH$_{EC}$ (n = 16), and HC (n = 18), using flow cytometry (Fig S3A). The distribution of CD4$^+$ and CD8$^+$ T cells, classical (CM), intermediate (IM), and non-classical monocytic (NCM) populations are depicted in UMAP (Fig 4A). The percentage of CD4$^+$ T cells were decreased, whereas CD8$^+$ T cells were increased in PLWH$_{EC}$ and PLWH$_{ART}$ compared with HC (Fig S3B). PLWH$_{EC}$ also exhibited a decreased proportion of CM compared with HC, but no other differences were identified on the monocytic subpopulations (Fig S3C). We did not observe any significant differences in ROS on CD4$^+$ or CD8$^+$ T cells (Figs 4B and S3D). In CM, ROS was significantly higher in PLWH$_{ART}$ samples compared with HC (Fig 4B). Some of the PLWH$_{ART}$ had higher ROS, whereas others expressed lower ROS on lymphocytic cell populations (Fig 4C). Therefore, to determine if long-term successfully treated HIV-1 infection influenced ROS production, we divided arbitrarily the PLWH$_{ART}$ group into long-term ART ([>10 yr,

n = 8] with a median of 19 [16–22] yr treatment) and short-term cART ([<10 yr, n = 8] with a median of 7 [6–8] yr treatment). Interestingly, levels of ROS were increased in long-term ART compared with short-term ART on CD4$^+$ T cells and compared with short-term ART and PLWH$_{EC}$ on CD8$^+$ T cells (Fig 4D). ROS levels were not affected by cART treatment on the monocytic cell populations (Fig S3E). These data highlights the effects of long-term cART treatment on oxidative stress and redox homeostasis in lymphocytic cell populations.

### Pharmacological inhibition of OXPHOS influences latency reversal and immunosenescence in an HIV latent lymphocytic cell model

In the ex vivo part of this study, we showed how up-regulation of OXPHOS was a signature of PLWH$_{ART}$ that differentiated them from the PLWH$_{EC}$ and how long-term treatment influenced oxidative stress and redox homeostasis on lymphocytic cell populations. Therefore, we decided to study the effect of inhibiting OXPHOS complexes I-V in a lymphocytic latency cell model (J-Lat 10.6) together with the parental cell line (Jurkat) by targeting complex I (metformin), complex II (D-α-tocopheryl succinate, aTOS), complex III (antimycin), complex IV (arsenic trioxide), and complex V (oligomycin) (Fig 5A) with respect to apoptotic properties, latency reversal and cellular senescence. The drugs did not have any effect on cell viability (Figs 5B and S4A), although inhibition of complex I, III, and IV in J-Lat 10.6 increased the levels of Annexin V (a marker of apoptosis) compared with the respective untreated control, whereas only inhibition of complex I and IV showed the same effect in Jurkat cells (Figs 5C and S4B). This indicates the role of OXPHOS complex III in the apoptotic properties of the HIV-1 latent cell model. It was only when inhibiting complex IV a significant increase in HIV-1 reactivation was observed in J-Lat 10.6 cells (Figs 5D and S4C). Several studies including ours have shown that PLWH$_{ART}$ has a potential for attenuated immune aging due to a shift in glutaminolysis, in a subset of PLWH$_{ART}$ who had dysregulated metabolic profiles. A recent pivotal study also indicated a role of glutaminolysis in senolysis (removal of senescence cells) as senescent cells are dependent on glutaminolysis (Johmura et al, 2021). To prove this, we measured the senescence markers CD57, Ki-67, and PCNA using flow cytometry and DNA damage marker H2A.X (S139) by Western blot. Cell surface expression of CD57 was not altered compared with the respective control when inhibiting the OXPHOS complexes although a baseline increase in CD57 was seen in J-Lat 10.6 cell compared with Jurkat (Fig S5A–C). The proportion of Ki-67–negative J-Lat 10.6 cells increased after inhibiting complexes II, III, and IV, whereas only inhibition of complex IV increased the proportion of Ki-67 negative Jurkat cells (Figs 5E and S5D). A mild decrease in KI-67 negative cells was also observed in J-Lat 10.6 cells when inhibiting complex I (Figs 5E and S5D). Inhibition of complex III increased PCNA negative J-Lat 10.6, whereas no significant differences were observed in Jurkat cells (Figs 5F and S5E). Furthermore, phosphorylation of H2A.X (S139) increased when inhibiting complex IV and decreased when inhibiting complex I, irrespective of the cell type (Figs 5G and H and S5F). The original blots were presented as a source file to Fig 5. Collectively, our data highlighted the potential role of pharmacological inhibition of the OXPHOS complexes with differential regulation of latency reversal, apoptotic properties, and

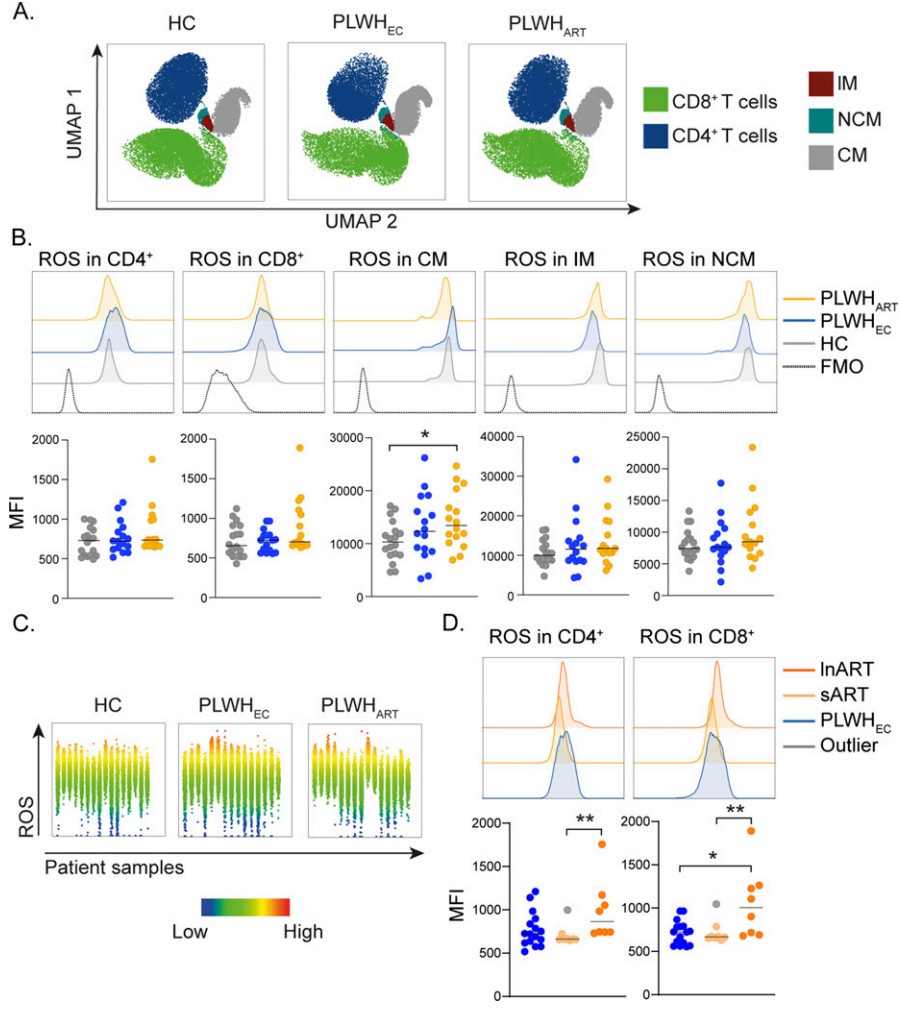

**Figure 4. Redox homeostasis during suppressive viremia.**
**(A)** Reactive oxygen species (ROS) detection in lymphocytic and monocytic cell populations from HC (n = 18), PLWH_EC (n = 16), and PLWH_ART (n = 18). UMAP representation showing the distribution of lymphocytic (CD4+ and CD8+ T cells) and monocytic (classical monocytes [CM], intermediate monocytes [IM], and non-classical monocytes [NCM]) cell populations. **(B)** Median fluorescence intensity (MFI) of ROS in CD4+, CD8+, CM, IM, and NCM in the cohort. Histograms show a representative sample from HC, PLWH_EC, and PLWH_ART exhibiting the median expression in each group. **(C)** Graphs showing MFI of ROS in each individual from HC, PLWH_EC, and PLWH_ART. **(D)** MFI of ROS in PLWH_EC (n = 16), short-term ART (sART, n = 8), and long-term ART (lnART, n = 8). Histograms show a representative sample from PLWH_EC, sART, and lnART exhibiting the median expression in each group. Statistical significance was determined using Mann–Whitney $U$ test ($P < 0.05$ with *<0.05, **<0.03, ***<0.002) and represented with median. See also Fig S3.

## Discussion

cellular senescence in lymphocytic HIV-1 latent cell model, depending on compounds and targeted complexes.

In the present study, we combined system-level blood cell transcriptomics and developed context-specific GSMM to provide a comprehensive system-level characterization of HIV-1 infected individuals with suppressive viremia either by natural (PLWH_EC) or drug-induced (PLWH_ART) control. The transcriptomic data identified up-regulation of OXPHOS as the characteristic feature of PLWH_ART, differentiating them from HIV-1 seropositive PLWH_EC, who were not on therapy. The main dysregulation seemed to occur in complexes I, III, and IV of the OXPHOS pathway. FBA identified altered flux in several glycolytic intermediates like pyruvate, αKG, glutamate, and fructose-6-phosphate in PLWH_ART compared with PLWH_EC and HC. Long-term cART also affected the redox homeostasis in T lymphocytes. The in vitro pharmacological inhibition of the OXPHOS complexes in the latent lymphocytic cell model suggested a role of the complex IV in latency reversal, complex I, III, and IV in apoptosis, and complex IV in immunosenescence.

Disrupted AA and central carbon metabolism have been proposed as a prominent characteristic of PLWH on long-term successful cART as reported by us and others (Mukerji et al, 2016; Babu et al, 2019; Rosado-Sánchez et al, 2019; Valle-Casuso et al, 2019; Gelpi et al, 2021; Meeder et al, 2021; Shytaj et al, 2021; Villumsen et al, 2022). Altered glutaminolysis (i.e., glutamine lysed to glutamate) and increased plasma glutamate have been observed in several cohorts from both high income (Gelpi et al, 2021) and low- and middle-income countries (Gelpi et al, 2021) and are required for optimal HIV-1 infection of CD4+ T cells (Clerc et al, 2019). Glutaminolysis is the primary pathway fueling the TCA cycle and OXPHOS in naïve and memory T cell subsets which are critical factors for immune recovery in successfully treated PLWH (Rosado-Sánchez et al, 2019). HIV-1 infection is more common in T cells with elevated glycolysis and OXPHOS and inhibition of these metabolic activities can block HIV-1 replication and reservoir transactivation (Valle-Casuso et al, 2019). Impairment of the metabolic steps preceding OXPHOS can also result in lipid accumulation in macrophages (Castellano et al, 2019). Enhanced glycolysis and OXPHOS are characteristics of CD8+ T cell exhaustion (Rahman et al, 2021). However, long-term molecular immune pathogenic consequences of successful cART have not yet been evaluated. In our study, we identified system-level

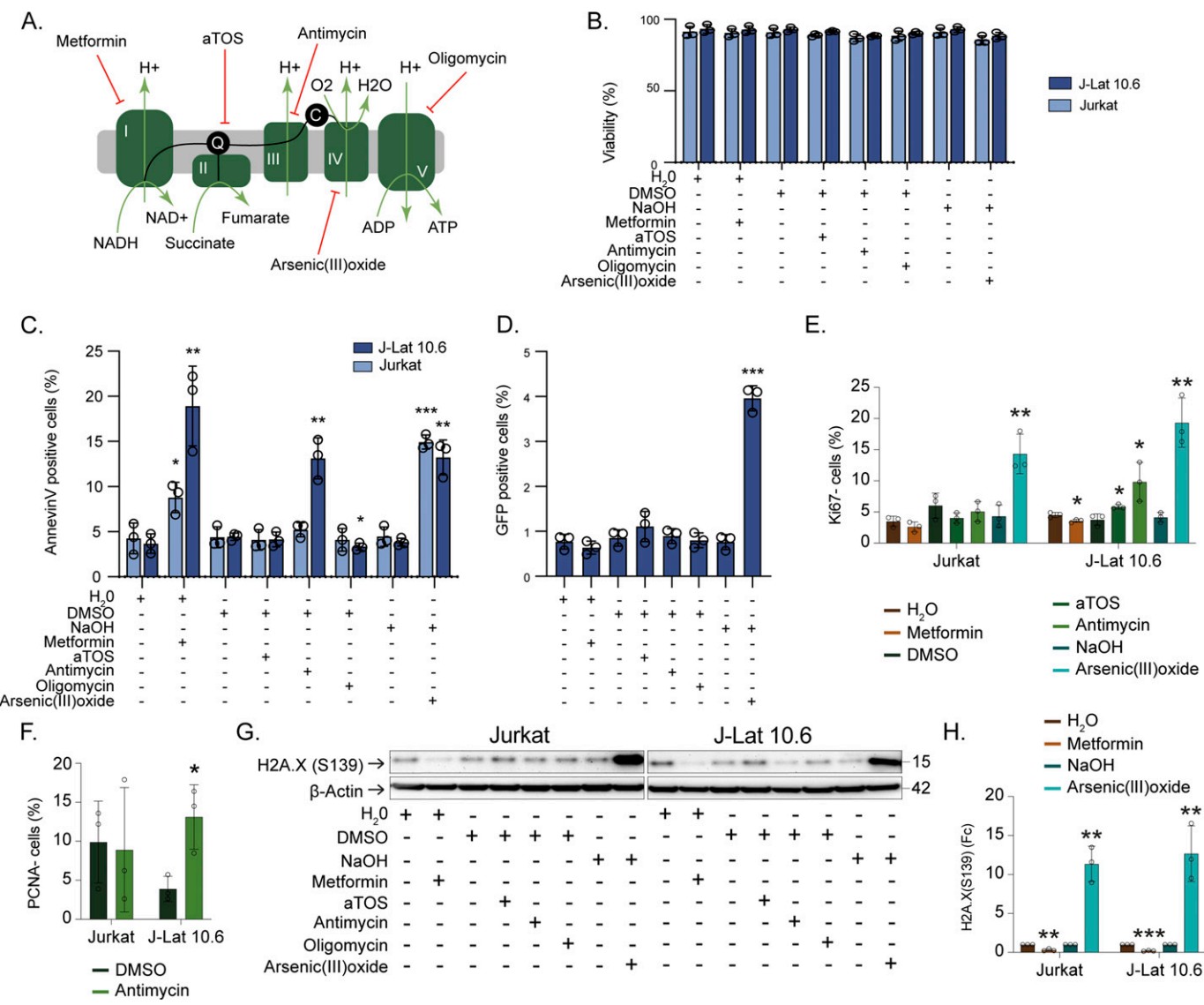

**Figure 5. Pharmacological inhibition of OXPHOS in lymphocytic HIV-1 latency cell model.**
**(A)** Schematic representation of inhibition of OXPHOS complexes with metformin (complex I), aTOS (complex II), antimycin (complex III), arsenic trioxide (complex IV), and oligomycin (complex V). **(B)** Drug toxicity for 24 h treatment of OXPHOS inhibitors. **(C)** Annexin V positive cells after treatment with OXPHOS inhibitors and respective controls. **(D)** Activation from latency in J-Lat 10.6 cells after treatment with OXPHOS inhibitors and respective controls. **(E)** Percentage Ki-67 negative cells after treatment with OXPHOS inhibitors or respective controls. **(F)** Percentage PCNA negative cells after treating with Antimycin or DMSO control. **(G)** Western blot detection of H2A.X (S139) and β-Actin in Jurkat and J-Lat 10.6 after treatment with OXPHOS inhibitors or respective controls. **(H)** Quantification of H2A.X (S139). The graph shows fold change (Fc) of protein expression in relation to respective control after normalization to β-Actin. Experiments were performed in three biological replicates. Significance was determined using two-tailed *t* test (*P* < 0.05 with * < 0.05, ** < 0.033, *** < 0.002) and represented with mean and SD. Significance for each drug is compared with respective control. See also Figs S4 and S5.
Source data are available online for this figure.

up-regulation of OXPHOS as the main characteristic of PLWH on long-term cART. When comparing with PLWH$_{EC}$, an up-regulation of OXPHOS, and to certain extent glycolysis, was observed in PLWH$_{ART}$. Given that HIV-1 preferentially selects cells that have elevated cellular OXPHOS and glycolysis for infection and replication, reservoir seeding (Hegedus et al, 2014; Palmer et al, 2014; Valle-Casuso et al, 2019) and cell-to-cell spread of HIV-1, this metabolic environment permit ongoing replication during cART (Sigal et al, 2011). Therefore, we hypothesize that up-regulation of OXPHOS in PLWH$_{ART}$ was the reason behind the relatively larger HIV-1 reservoir

in long-term successfully treated infection compared with PLWH$_{EC}$ with natural control of viral replication. This metabolic modulation could potentially be a barrier to the post-treatment control of viral replication.

A recent seminal study showed that a higher HIV-1 viral set point in untreated patients during acute HIV-1 infection correlated positively with OXPHOS and that in vitro pharmacological inhibition of complex I (by rotenone or metformin) and complex III (by antimycin A) suppressed viral replication and immunometabolism through an NLRX1 and FASTKD5-dependent mechanism (Guo et al,

2021). Furthermore, we recently observed that blocking glycolysis with 2-deoxyglucose (2-DG) increased cell death in lymphocytic and pre-monocytic HIV-1 latent cell models (Gelpi et al, 2021), in line with other studies (Valle-Casuso et al, 2019; Guo et al, 2021). These studies indicate a critical role for glycolysis and OXPHOS in HIV-1 immuno-pathogenesis. In the present study, we observed that latently infected cells treated with antimycin resulted in increased markers of apoptosis in latent J-Lat 10.6 cells compared with the parental Jurkat cells. This indicates an increased preferential cell death of the latently infected cells without latency reversal. Only inhibition of complex IV by arsenic trioxide showed a small degree of latency reversal. Both the transcriptomic data and the in vitro assays indicated the role of complex I, III, and IV as essential components of the electron transport chain for generation of ATP and cellular energy requirements. Complexes I and III have a role in ROS production and are essential in inflammatory macrophages and T helper 17 ($T_H$17) cells while also playing a vital role in lymphocyte activation, proliferation, and differentiation (Yin & O'Neill, 2021). Recently, it has been shown that complex III is crucial for the suppressive function of Tregs (Weinberg et al, 2019). Our DCQ identified increased frequency of Tregs in PLWH$_{ART}$ compared with PLWH$_{EC}$ which is in line with recent findings (Caetano et al, 2020); however, PLWH$_{EC}$ can present more activated Tregs (Gaardbo et al, 2014; Caetano et al, 2020). Finally, an earlier study reported the Cox-II enzyme leads to reduced T-cell apoptosis in HIV-1 infected cells (Tripathy & Mitra, 2010). In contrast, our study indicated pharmacological inhibition of the complex IV with arsenic trioxide increased apoptosis (as measured by the annexin V) both in latent J-Lat 10.6 cells and non-latent Jurkat. Interestingly, inhibition of complex IV in J-Lat 10.6 cells also showed latency reactivation which could potentially be linked to apoptosis.

In our FBA, we identified altered flux in pyruvate, glutamate, and $\alpha$KG in the PLWH$_{ART}$ compared with PLWH$_{EC}$ and HC. Recently, we identified a higher level of glutamate in PLWH$_{ART}$ in several cohorts compared with HC (Gelpi et al, 2021). The level was even higher in PLWH$_{ART}$ with metabolic syndrome (Gelpi et al, 2021). Blood glutamate level has been reported to be higher in PLWH with dementia (Ferrarese et al, 2001). Reducing the blood glutamate concentrations with blood glutamate scavengers like pyruvate facilitates the efflux of glutamate from the brain to the blood. This can limit the neurotoxic effect of glutamate (Boyko et al, 2012) and has been reported to effectively improve neurological recovery in traumatic brain injury (Gottlieb et al, 2003; Zlotnik et al, 2007; Boyko et al, 2012). The coordination between glutamate and pyruvate and its neuroprotective role in chronic HIV-1 infected patients on therapy needs further studies to understand neurological complications in HIV infection after successful treatment.

Although immune cell senescence decreases the overall cellular activity, it is associated with a high metabolic need, usually by increasing aerobic glycolysis. In the case of our lymphocytic cell culture model, we detected an enrichment of the senescent marker CD57 compared with the parental cell line, indicative of increased chronic activation of latently infected cells. Furthermore, we detected increased levels of DNA damage (H2A.X [S139]) (Mah et al, 2010), decreased proliferation (Ki-67) (Lawless et al, 2010), and DNA replication (PCNA) (González-Magaña & Blanco, 2020) after OXPHOS inhibition. Earlier studies have shown how OXPHOS inhibition in

human fibroblasts induced senescence (Stöckl et al, 2006). Therefore, high plasticity of metabolic reprogramming could induce an increase in glycolysis during OXPHOS inhibition which could potentially be coupled to induction of senescence in the HIV-1 latent cells during the suppressive therapy.

Our study also showed that ROS was increased in patients on long-term (median 19 yr) compared with short-term (median 7 yr) of suppressive therapy. This could be linked to the use of the older nucleoside reverse transcriptase inhibitors (NRTIs) like zidovudine (AZT), stavudine (d4T), or didanosine (ddI) as a part of the initial treatment regimen. The cell's epigenetic state is closely associated with ROS-induced oxidative stress due to mitochondrial damage and altered OXPHOS (Guillaumet-Adkins et al, 2017). It is known that antiretrovirals such as AZT, d4T, and ddI can cause mitochondrial damage, ultimately altering OXPHOS (Pinti et al, 2006). Recent molecular studies have reported that PLWH on treatment has epigenetic age acceleration (Gross et al, 2016) compared with the non-infected individuals that can partially be reversed with cART initiation (Esteban-Cantos et al, 2021). Therefore, understanding the biological mechanism of potential accentuated aging in PLWH on long-term successful therapy who were exposed to earlier generation treatment regimen and dysregulated metabolic profile could potentially provide a clinical intervention strategy to improve the quality of life of PLWH$_{ART}$.

In conclusion, our study indicated a system-level up-regulation of OXPHOS and, to a certain extent, glycolysis in PLWH$_{ART}$ compared with the PLWH$_{EC}$. Furthermore, we show how this up-regulation could play a role in latent reservoir dynamics and immunosenescence in HIV-1–infected individuals with long-term successful therapy. Pharmacological inhibition of the OXPHOS complexes could have a role in latency reversal, apoptotic properties, and immunosenescence in latently infected cells. Further studies are warranted to elucidate the molecular mechanisms underlying the observed shift in OXPHOS in PLWH$_{ART}$ and how its coordination with glutaminolysis can lead to immune dysregulation during successful therapy.

## Materials and Methods

### Cohort description

The study population includes three groups of PLWH, with two groups as suppressed viremia (PLWH$_{ART}$ and PLWH$_{EC}$, n = 19 each), and one group with viremia (PLWH$_{VP}$ herein, n = 19). In addition, we enrolled 19 HC. The study was approved by the regional ethics committees of Stockholm (2013/1944-31/4 and 2009/1485-31) and amendment (2019-05585 and 2019-05584, respectively) and performed in accordance with the Declaration of Helsinki. All participants gave informed consent. The patient's identity was anonymized and delinked before analysis.

### Transcriptomics analysis

PBMCs were used for RNA-sequencing (RNA-Seq) using Illumina HiSeq2500 or NovaSeq6000 as described by us (Akusjärvi et al, 2022). Differential gene expression analysis was performed using

the R/Bioconductor package DESeq2 v1.26.0 (DOI: 10.18129/B9.bioc.DESeq2). Gene list enrichment analysis for cART-specific genes was performed using enrichr module of python package GSEAPY v0.9.16 (Subramanian et al, 2005; Chen et al, 2013) and MSigDB hallmark gene set v7.4. GSEA between $PLWH_{ART}$ and $PLWH_{EC}$ was performed using GSEA v4.1.0 software (Subramanian et al, 2005) and MSigDB hallmark gene set v7.4. Metabolomics data were generated using the Metabolon HD4 (Metabolon Inc.) (Akusjärvi et al, 2022).

## DCQ

DCQ by measuring the proportion of different cells in each sample was performed using the deconvolution algorithm adapted from Estimating the Proportions of Immune and Cancer cells (Chen et al, 2013). The reference gene expression profile consists of gene-level expression data of 18 blood cell types and it is based on Human Protein Atlas version 20.1 and Ensembl version 92.38. Signature genes for the 18 blood cell types in the reference profile were downloaded from CellMarker (Zhang et al, 2019) and PanglaoDB (Franzén et al, 2019). The transcript per million (TPM) transformed gene expression data of all genes from the samples were used in the procedure along with reference profile and signature gene list to estimate the cell proportion.

## ART-specific gene identification

Significantly regulated genes (adjusted $P < 0.05$) in all the pair-wise comparisons among the four cohorts were used to derive the cART-specific genes. The list of significant genes in each of the comparisons was considered as individual sets and various set operation procedures were used for the derivation. The set operations performed are represented below.

$$ART = \{z \mid z \in X_1 \text{ or } z \in X_2 \text{ or } z \in X_3\}$$
$$NULL = \{z \mid z \in Y_1 \text{ or } z \in Y_2 \text{ or } z \in Y_3\}$$
$$ART\text{-specific genes} = \{z \mid z \in ART \text{ and } z \notin NULL\}$$

where,

$X_1 = \{z \mid z$ is gene regulated in HC versus $PLWH_{ART}\}$
$X_2 = \{z \mid z$ is gene regulated in $PLWH_{EC}$ versus $PLWH_{ART}\}$
$X_3 = \{z \mid z$ is gene regulated in $PLWH_{VP}$ versus $PLWH_{ART}\}$
$Y_1 = \{z \mid z$ is gene regulated in HC versus $PLWH_{VP}\}$
$Y_2 = \{z \mid z$ is gene regulated in $PLWH_{EC}$ versus $PLWH_{VP}\}$
$Y_3 = \{z \mid z$ is gene regulated in HC versus $PLWH_{EC}\}$

## GSMM, FBA, and essentiality analysis

Group-specific human GSMMs were reconstructed by integrating transcriptomics data on human reference GSMM obtained from Metabolic Atlas (Robinson et al, 2020). The metabolic model reconstruction was performed using task-driven Integrative Network Inference for Tissues (tINIT) algorithm (Agren et al, 2012, 2014; Robinson et al, 2020). The algorithm creates a context-specific model by selecting only reaction that can carry flux based on the provided transcript expression table (transcript per million). The

reconstructed models were then checked for biological feasibility by analyzing their capacity to carry out 56 essential metabolic tasks. FBA was performed using MatLab function solveLP from RAVEN toolbox v2.4.0 (Wang et al, 2018) and ATP hydrolysis as objective function. Plasma metabolomics data were used as a reference to constrain the exchange reactions in the model assuming that exchange reaction fluxes were relatively influenced by availability of extracellular metabolites. We calculated $log_2$-scaled changes of exchange metabolites against the control cohort, and it was used proportionally to compute the reaction bounds.

Network topology analysis was performed on the metabolic networks generated for the cohorts. The metabolic networks were created by drawing edges between reactants, products, and enzymatic genes of each of the reactions, which showed significant ($>10^{-5}$) and varying flux values among the cohorts. The networks were then analyzed using igraph toolkit. The absolute value of the flux scaled between 0 and 1 was used as edge weight. Leiden algorithm (Traag et al, 2019) was used to identify communities and the betweenness centrality of all the nodes was computed. Nodes were ranked based on their centrality measurement. Nodes with high centrality were considered as most influential for the existence and functioning of the network.

## Visualization

R package ggplot2 v3.3.2 (Wickham, 2016) was used to create all bubble plots, scatter plots, and boxplots. R/Bioconductor package ComplexHeatmap v2.2.0 (Gu et al, 2016) was used to create all the heat maps. Network diagrams were drawn in Cytoscape ver 3.6.1 (Shannon et al, 2003). Venn diagrams were generated using the online tool InteractiVenn (Chen & Boutros, 2011).

## Total HIV DNA quantification

Total DNA was extracted from PBMCs using QIAamp DNA mini kit (QIAGEN) according to manufacturers' instructions. HIV-1 DNA quantification was performed using Internally Controlled qPCR (IC-qPCR) as described by Vicenti et al (2018). In brief, total HIV-1 DNA was quantified in $PLWH_{ART}$ (n = 17) and $PLWH_{EC}$ (n = 14) using 500 ng of DNA in duplicates. Quantification was performed using Takara Universal Mastermix (Takara) on an ABI 7500F using primers ($\beta$ globin F; AGGGCCTCACCACCAACTT, $\beta$ globin R; GCACCTGACTCCTGAGGAGAA, HXB2 F; GCCTCAATAAAGCTTGCCTTGA, HXB2 R; GGCGCCACTGCTAGA-GATTTT) and probes ($\beta$ globin; HEX-ATCCACGTTCACCTTGCCCCACA-TAM, HXB2; FAM-AAGTAGTGTGTGCCCGTCTG-MGBEQ) targeting $\beta$ globin and HIV-1 (HXB2) and normalized to $\beta$ globin levels.

## Cell culture

The latency cell model J-Lat clone 10.6 (NIH HIV reagent program) was used together with its parental cell line Jurkat. Cells were cultured in StableCell RPMI 1640 (Sigma-Aldrich) supplemented with 10% fetal bovine serum (Gibco) and 20 U/ml penicillin and 20 $\mu$g/ml streptomycin (Gibco) at 37°C and 5% $CO_2$.

## Drug treatment

Cytotoxicity assays were performed for metformin (Sigma-Aldrich), arsenic trioxide (Sigma-Aldrich), oligomycin (Sigma-Aldrich), antimycin (Sigma-Aldrich), and aTOS (Sigma-Aldrich) (Fig S3A). Experimental concentrations with low cytotoxicity were chosen and assayed for 24 h. All assays were performed in biological triplicates and analyzed for viability using flow cytometry, as described below.

## Flow cytometry

PBMCs were subjected to flow cytometry analysis. Samples were thawed in 37°C water bath and washed with flow cytometry buffer (PBS + 2% FBS + 2 mM EDTA). Total cellular ROS levels were measured using the CellROX Deep Red Flow Cytometry Assay Kit (Invitrogen) according to the manufacturer's instructions. Briefly, 750 nM of CellROX deep red reagent was added to PBMCs and incubated for 1 h at 37°C, protected from light. The cells were then stained with Live/Dead fixable near-IR dye (Invitrogen), and cell surface markers were detected by incubating cells with anti-CD3 (clone OKT3, BD Bioscience), anti-CD4 (clone SK3; BD Bioscience), anti-CD8 (clone HIT8a; BioLegend), anti-CD14 (clone M5E2; BioLegend), and anti-CD16 (clone 3G8; BD Bioscience) for 20 min on ice in flow cytometry buffer. Positive and negative controls for ROS measurement were performed by incubating cells with either tert-butyl hydroperoxide (200 $\mu$M) or N-acetyl cysteine (5 mM) for 45 min at 37°C before the addition of CellROX deep red reagent. All cells were fixed with 2% paraformaldehyde before acquiring on BD FACS Symphony flow cytometer (BD Bioscience). Compensation setup was performed using single-stained controls prepared with antibody-capture beads: anti-mouse Ig, $\kappa$/negative control compensation particles set (BD Biosciences) for mouse antibodies and ArC amine-reactive compensation bead kit (Invitrogen) for use with LIVE/DEAD fixable dead cell stain kits.

Flow cytometry for cell lines was conducted by extracellular staining using anti-CD57 (clone HNK-1; BioLegend) and LIVE/DEAD Near-IR viability stain (Invitrogen) followed by fixation using ki-67 fixation/permeabilization kit (eBioscience). Intracellular staining was performed using anti-Ki-67 (clone Ki-67; BioLegend) and anti-PCNA (clone PC10; BioLegend). Analysis of apoptosis was performed using Annexin-V Alexa647 conjugate (Thermo Fisher Scientific) staining in combination with LIVE/DEAD Near-IR viability stain (Invitrogen) prior fixation using 4% PFA. Samples were acquired on BD FACS Fortessa (BD Bioscience). Flow cytometry data were analyzed and compensated with FlowJo 10.6.2 (TreeStar Inc.) and statistical analysis was performed using Mann–Whitney *U* test or two-tailed *t* test in Prism 9.3.0 (GraphPad Software Inc.).

## Western blot

Cells were lysed in RIPA buffer (Sigma-Aldrich) supplemented with 1× PhosSTOP (Sigma-Aldrich) and 2× cOmplete protease inhibitor cocktail (Roche) on ice for 30 min. Protein estimation was performed using DC protein assay (Bio-Rad Laboratories) and 37.5–48 $\mu$L of protein run in each well on NuPage 4–12% BisTris 20 well, 1 mm precast gels (Thermo Fisher Scientific) and transferred using the iBlot transfer system (Invitrogen) with iBlot PVDF Transfer stack (Invitrogen). Membranes were incubated with Phospho-Histone H2A.X (Ser139) (Cell Signaling Technology) and $\beta$-Actin (Sigma-Aldrich). Secondary antibody incubation was performed using Dako Immunoglobulins/HRP (Aglient Technologies) and membranes developed using ECL (Amersham) on ChemiDoc (Bio-Rad Laboratories). Relative protein quantification was performed using ImageLab 6.0.1 (Bio-Rad Laboratories) and statistical significance using a two-tailed *t* test in Prism 9.3.0 (GraphPad Software Inc.).

# Data Availability

The raw RNA sequencing (RNAseq) data have been deposited in the NCBI/SRA with PRJNA420459. The metabolomics data are available from dx.doi.org/10.6084/m9.figshare.19747582. All the codes are available at github: github.com/neogilab/LongART.

# Supplementary Information

# Acknowledgements

The authors acknowledge support from the National Genomics Infrastructure in Genomics Production Stockholm funded by Science for Life Laboratory, the Knut and Alice Wallenberg Foundation and the Swedish Research Council, and SNIC/Uppsala Multidisciplinary Center for Advanced Computational Science for assistance with massively parallel sequencing and access to the UPPMAX computational infrastructure. The study is funded by the Swedish Research Council grants 2017-01330, 2018-06156, and 2021-01756 to U Neogi and 2017-05848 and 2020-02129 to A Sönnerborg.

## Author Contributions

AT Ambikan: data curation, formal analysis, investigation, visualization, methodology, and writing—original draft.
S Svensson-Akusjärvi: data curation, formal analysis, visualization, methodology, and writing—original draft.
S Krishnan: visualization, methodology, and writing—review and editing.
M Sperk: formal analysis, methodology, and writing—review and editing.
P Nowak: data curation, investigation, and writing—review and editing.
J Vesterbacka: data curation, investigation, and writing—review and editing.
A Sönnerborg: conceptualization, resources, funding acquisition, investigation, project administration, and writing—review and editing.
R Benfeitas: supervision, visualization, methodology, project administration, and writing—review and editing.
U Neogi: conceptualization, resources, supervision, funding acquisition, visualization, methodology, writing—original draft and project administration.

**Life Science Alliance**

## Conflict of Interest Statement

The authors declare that they have no conflict of interest.

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
