## [Reviewer comments · Life Science Alliance]

Life Science Alliance

Genome-scale metabolic models for natural and long-term drug-induced viral control in HIV-infection

Anoop Ambikan, Sara Svensson-Akusjärvi, Shuba Krishnan, Maike Sperk, Piotr Nowak, Jan Vesterbacka, Anders Sönnberg, Rui Benfeitas, and Ujjwal Neogi

DOI: <https://doi.org/10.26508/lsa.202201405>

Corresponding author(s): Ujjwal Neogi, Karolinska Institute

Review Timeline:

Submission Date:	2022-02-11
Editorial Decision:	2022-03-20
Revision Received:	2022-03-21
Editorial Decision:	2022-04-28
Revision Received:	2022-05-02
Accepted:	2022-05-02

Scientific Editor: Novella Guidi

Transaction Report:

March 20, 2022

Re: Life Science Alliance manuscript #LSA-2022-01405-T

Ujjwal Neogi
Karolinska Institute
LabMed
ANA Futura
ANA8 Plan 7
Huddinge, Stockholm 14152
Sweden

Dear Dr. Neogi,

Thank you for submitting your manuscript entitled "Genome-scale metabolic models for natural and long-term drug-induced viral control in HIV-infection" to Life Science Alliance. The manuscript was assessed by expert reviewers, whose comments are appended to this letter. We, thus, encourage you to submit a revised version of the manuscript back to LSA that responds to all of the reviewers' points.

Thank you for this interesting contribution to Life Science Alliance. We are looking forward to receiving your revised manuscript.

Sincerely,

B. MANUSCRIPT ORGANIZATION AND FORMATTING:

Reviewer #1 (Comments to the Authors (Required)):

The work by Anoop et al. concerns explorative analyses and modelling on multi-omic profiling data of subjects affected by HIV. The investigation aims at the characterization of molecular mechanisms specifically triggered by combined antiretroviral therapy (ART) by comparison of treated subjects, "elite controllers", healthy subjects. Explorative analyses based on systems biology approaches are accompanied by pharmacological inhibition of key molecular elements found as results. Overall, the research confirms previous observations but adds further relevant details.

Albeit interesting some limitations prevent the publication in the present form.

Fig.1C reports the outcome of enrichment analysis on specifically dysregulated genes in PLWHart. Only adjusted significance should be indicated. In the description of methods the Gene Set Enrichment Analysis is indicated as the tool used to obtain these results. However, this is an over-representation analysis of annotations (results are not obtained by GSEA application). The heatmap depicted in FIG.2C is not used properly. The pathway analysis reported in FIG.2D is obtained by GSEA, thus the correspondent heatmap should be used to describe the results. Moreover, in the previous analysis, to show the informativeness of the found signature a UMAP diagram was used.

It is not clear the overlap between the two set of genes identified (specifically deregulated in PLWHart and differentially expressed between PLWHart and PLWHec). This part should be improved.

As concerns the involvement of OXPHOS complexes, how many and which genes are specifically differentially expressed in the comparison PLWHart vs PLWHec?

Fig.S2 reports a the heatmap of dysregulated genes but here unsupervised clustering of samples does not add/provide specific information.

In the inhibition experiments, all the complexes have been targeted: they are not focused only on complexes I, III, IV. Some explanations could be provided.

minor comments

Table S1 is indicated as Table 1 in supplemental material.

p.3 it is not clear the meaning of "proteo-transcriptomic" in the description.

p.5: "T-regs" should be written explicitly at the first time it is mentioned.

FIG. 2D values (0.25 0.5 ...) indicated in the legend are not coherent with the transformation reported ($-\log_{10}()$)

Reviewer #2 (Comments to the Authors (Required)):

Ambikan et al. presented a study on "Genome-scale metabolic models for natural and long-term drug-induced viral control in HIV-infection". Using transcriptomic information from PBMCs of a cohort of untreated HIV patients, HIV patients under ART, elite controllers and healthy controls, the authors performed a comparative metabolic network/flux balance analysis to identify key metabolic processes (such as oxidative phosphorylation and glycolysis) with differential flux balance between the different groups of the cohort. The authors further used a cell line platform to investigate the effect of interrupted oxidative phosphorylation on latency. This is an innovative study with highly relevant results.

Main points:

System-level PBMC-based gene expression identifies up-regulation of OXPHOS in PLWH(ART):

The presented data (in the figures) are not completely supportive. The authors should emphasize the "up-regulation of OXPHOS in PLWH(ART)" in Fig. 1.

Fig 1C: For each enriched pathway, I presume, the depicted nodes refer to all overlapping genes between expressed genes and genes in the particular pathway. The non-overlapping genes with high mean expression (large nodes) should be labeled.

Larger viral reservoir and up regulated OXPHOS differentiate PLWH(ART) from PLWH(EC):

The data is strongly supportive of this claim.

Altered flux balance in PLWH on cART is linked to OXPHOS, glycolysis, and TCA cycle:

The data is somewhat supportive of this claim. However, the authors should elaborate further details on their flux balance approach (as the Matlab files including detailed flux balance models are not available to the reviewer). They should provide further details how they initially selected the initial gene set that yield about ~1800 genes/transcripts in their models.

The authors should also elaborate how they addressed missing but required reactions and if (and how) where associated (if possible) with corresponding enzymes. Specifically, in the case these orphan reactions change between the different HIV groups, the authors should discuss these findings and provide an interpretation. Furthermore, the authors may want to explore methods to identify differential connectivity to identify network differences between the different flux balance based modules shown in Fig. 3D.

Long term cART disrupts redox homeostasis in the lymphocytic cell population:

The data is strongly supportive of this claim.

Pharmacological inhibition of OXPHOS influences latency reversal and immunosenescence in an HIV latent lymphocytic cell model:

The data is strongly supportive of this claim.

Additional issues:

Figs. 2C, 2E and S2: the color scale of Z-scores should be identical. Using different colors for the same Z-value is irritating.

The supplementary files 1 - 5 are not available.

Reviewer #1 (Comments to the Authors (Required)):

The work by Anoop et al. concerns explorative analyses and modelling on multi-omic profiling data of subjects affected by HIV. The investigation aims at the characterization of molecular mechanisms specifically triggered by combined antiretroviral therapy (ART) by comparison of treated subjects, "elite controllers", healthy subjects. Explorative analyses based on systems biology approaches are accompanied by pharmacological inhibition of key molecular elements found as results. Overall, the research confirms previous observations but adds further relevant details. Albeit interesting some limitations prevent the publication in the present form.

Reply: We are thankful to the reviewer for the positive evaluation of our manuscript.

Fig.1C reports the outcome of enrichment analysis on specifically dysregulated genes in PLWHart. Only adjusted significance should be indicated. In the description of methods, the Gene Set Enrichment Analysis is indicated as the tool used to obtain these results. However, this is an over-representation analysis of annotations (results are not obtained by GSEA application).

Reply: We are thankful for the suggestion and agree with the reviewer. We have now removed the unadjusted p-values. We also changed the term of the analysis in Fig 1C to gene list enrichment analysis that used by the enrichr software.

The heatmap depicted in FIG.2C is not used properly. The pathway analysis reported in FIG.2D is obtained by GSEA, thus the correspondent heatmap should be used to describe the results. Moreover, in the previous analysis, to show the informativeness of the found signature a UMAP diagram was used. It is not clear the overlap between the two set of genes identified (specifically deregulated in PLWHart and differentially expressed between PLWHart and PLWHec). This part should be improved.

Reply: We are thankful for the comment. The UMAP was for ART-specific genes. While fig 2 is a comparative analysis between PLWH_{ART} and PLWH_{EC}. However it is an excellent suggestion to check the overlap between the two sets of genes i.e. cART specific genes and DGE between PLWH_{ART} and PLWH_{EC}. We performed the analysis and it turned out that the common genes mainly belonged to the OXPPOS and ROS signaling further strengthened our result.

We mentioned in the text as follows:

"We also checked the overlap between the ART specific genes (n=1037) and the differentially regulated genes between the PLWH_{ART} and PLWH_{EC} (n=1061). We observed 557 genes overlapped between the two sets of the genes. The gene list enrichment analysis using MSigDB hallmark gene-set identified OXPPOS (Adjusted p<0.001), MYC targets V1 (Adjusted p=0.004) and ROS pathway (Adjusted p=0.04) were significant pathways."

As concerns the involvement of OXPPOS complexes, how many and which genes are specifically differentially expressed in the comparison PLWHart vs PLWHec? Fig.S2 reports a the heatmap of dysregulated genes but here unsupervised clustering of samples does not add/provide specific information.

Response to Reviewer

Reply: We are thankful for the suggestion we have now revised fig 2E and added the genes that are significantly differs between PLWH_{ART} and PLWH_{EC}. WE also mentioned the proportions and number of the genes that significantly differ between the groups in different complexes as follows.

“Therefore, we looked at OXPPOS in detail to find which complexes were most affected. Among the genes in the five complexes of OXPPOS (I to V), complexes I (34%), III (45%), and IV (45%) were primarily affected in PLWH_{ART} compared to PLWH_{EC} (Fig 2E).”

In the inhibition experiments, all the complexes have been targeted: they are not focused only on complexes I, III, IV. Some explanations could be provided.

Reply: We would like to thank the reviewer for pointing this out. We used pharmacological inhibition to study the effect of OXPPOS complexes as they are dysregulated between PLWH_{ART} and PLWH_{EC}. In Fig 5, we presented all the complex data where there were differences. We mentioned that mainly OXPPOS complex I, III, and IV was affected to synchronize the patient data. However, there were no difference in complex II and only difference between Jurkat and J-lat were annexin-V in complex V. Additionally, to prove that one or some of the complexes are affected during latency, we had to show that it was not just a consequence of general inhibition of OXPPOS. Therefore, we picked drugs known from the cancer field, each targeting one of the OXPPOS complexes to show that it was a true effect of that complex.

minor comments

Table S1 is indicated as Table 1 in supplemental material.

Reply: Thank you for pointing out this. We have now modified.

p.3 it is not clear the meaning of "proteo-transcriptomic" in the description.

Reply: We have now changed it to “Moreover, **integrative proteomics and transcriptomic**”

p.5: "T-regs" should be written explicitly at the first time it is mentioned.

Reply: We have now elaborated it.

FIG. 2D values (0.25 0.5 ...) indicated in the legend are not coherent with the transformation reported (-log₁₀())

Reply: We are thankful for the critical checking of our manuscript. We have cross-checked this. It is correct. We used the false discovery rate (FDR: as provided by the tool). However we noticed an error in the text. We reported nominal p values (also provided by the tool) which we have now replaced with FDR and throughout the manuscript and figure legends in concordance with other analyses. We used multiple hypothesis corrections throughout the manuscript.

Response to Reviewer

Reviewer #2 (Comments to the Authors (Required)):

Ambikan et al. presented a study on "Genome-scale metabolic models for natural and long-term drug-induced viral control in HIV-infection". Using transcriptomic information from PBMCs of a cohort of untreated HIV patients, HIV patients under ART, elite controllers and healthy controls, the authors performed a comparative metabolic network/flux balance analysis to identify key metabolic processes (such as oxidative phosphorylation and glycolysis) with differential flux balance between the different groups of the cohort. The authors further used a cell line platform to investigate the effect of interrupted oxidative phosphorylation on latency. This is an innovative study with highly relevant results.

Reply: We are thankful to the reviewer for the positive evaluation of our manuscript.

Main points:

System-level PBMC-based gene expression identifies up-regulation of OXPHOS in PLWH(ART): The presented data (in the figures) are not completely supportive. The authors should emphasize the "up-regulation of OXPHOS in PLWH(ART)" in Fig. 1. Fig 1C: For each enriched pathway, I presume, the depicted nodes refer to all overlapping genes between expressed genes and genes in the particular pathway. The non-overlapping genes with high mean expression (large nodes) should be labeled.

Reply: We are thankful for the suggestion. We agree with the reviewer the upregulation of OXPHOS is in Fig 2. Therefore we changed the title "System-level PBMC-based gene expression identifies **dysregulation** of OXPHOS in PLWH_{ART}" and replaced the upregulation with dysregulation. We also labeled the genes with high expression in Fig 1C.

Larger viral reservoir and up regulated OXPHOS differentiate PLWH(ART) from PLWH(EC): The data is strongly supportive of this claim.

Reply: We are thankful to the reviewer for the positive evaluation of our manuscript.

Altered flux balance in PLWH on cART is linked to OXPHOS, glycolysis, and TCA cycle: The data is somewhat supportive of this claim. However, the authors should elaborate further details on their flux balance approach (as the Matlab files including detailed flux balance models are not available to the reviewer). They should provide further details how they initially selected the initial gene set that yield about ~1800 genes/transcripts in their models. The authors should also elaborate how they addressed missing but required reactions and if (and how) where associated (if possible) with corresponding enzymes. Specifically, in the case these orphan reactions change between the different HIV groups, the authors should discuss these findings and provide an interpretation. Furthermore, the authors may want to explore methods to identify differential connectivity to identify network differences between the different flux balance based modules shown in Fig. 3D.

Response to Reviewer

Reply: Thank you for your comments. We have used tINIT algorithm to reconstruct the group (context) specific genome-scale metabolic models from human reference model by inputting the mean gene expression table corresponding each of the groups. The algorithm only selects reactions that can carry flux based on the gene expression values provided. We further confirmed the biological feasibility of the reconstructed model by checking their capability of performing 56 essential metabolic tasks known to occur in all cell types. The method is described in detail by us in another paper <http://dx.doi.org/10.2139/ssrn.3988390>. However, we have now briefly described the method in the manuscript. In the first version we were not able to upload the matlab file. We have now given that in the supplementary zip file.

We have also checked if any cohort-specific network module exists at functional level by performing metabolite set enrichment analysis of all modules shown in Fig 3D and the overlap among them. The analysis did not identify any unique modules. Thus we have not reported it. The main objective of the metabolic network topology analysis to identify metabolically important genes and metabolites by ranking them based on their influence on the network.

The following text has been added:

The metabolic model reconstruction was performed using task-driven Integrative Network Inference for Tissues (tINIT) algorithm (Agren, Bordel et al., 2012, Agren, Mardinoglu et al., 2014, Robinson et al., 2020). The algorithm creates a context-specific model by selecting only reaction that can carry flux based on the provided transcript expression table (TPM). The reconstructed models were then checked for biological feasibility by analyzing their capacity to carry out 56 essential metabolic tasks.

Long term cART disrupts redox homeostasis in the lymphocytic cell population:
The data is strongly supportive of this claim.

Pharmacological inhibition of OXPHOS influences latency reversal and immunosenescence in an HIV latent lymphocytic cell model:

The data is strongly supportive of this claim.

Reply: We are thankful to the reviewer for the positive evaluation of our manuscript.

Additional issues:

Figs. 2C, 2E and S2: the color scale of Z-scores should be identical. Using different colors for the same Z-value is irritating.

Reply: We have now changed it as suggested.

The supplementary files 1 - 5 are not available.

Reply: We have now provided it as zip files as we could not upload the matlab files.

April 28, 2022

RE: Life Science Alliance Manuscript #LSA-2022-01405-TR

Dr. Ujjwal Neogi
Karolinska Institute
LabMed
ANA Futura
ANA8 Plan 7
Huddinge, Stockholm 14152
Sweden

Dear Dr. Neogi,

Thank you for submitting your revised manuscript entitled "Genome-scale metabolic models for natural and long-term drug-induced viral control in HIV-infection". We would be happy to publish your paper in Life Science Alliance pending final revisions necessary to meet our formatting guidelines.

-please upload your supplementary files and tables as single files; the tables should be uploaded in editable doc or excel format

Figure Issues:

-Figure S6 should be labeled as source data rather than a supplementary figure since this is a source data file for figure 5

A. FINAL FILES:

B. MANUSCRIPT ORGANIZATION AND FORMATTING:

Sincerely,

Reviewer #2 (Comments to the Authors (Required)):

All my concerns have been addressed by the authors.

May 2, 2022

RE: Life Science Alliance Manuscript #LSA-2022-01405-TRR

Dr. Ujjwal Neogi
Karolinska Institute
LabMed
ANA Futura
ANA8 Plan 7
Huddinge, Stockholm 14152
Sweden

Dear Dr. Neogi,

Thank you for submitting your Research Article entitled "Genome-scale metabolic models for natural and long-term drug-induced viral control in HIV-infection". It is a pleasure to let you know that your manuscript is now accepted for publication in Life Science Alliance. Congratulations on this interesting work.

DISTRIBUTION OF MATERIALS:

Again, congratulations on a very nice paper. I hope you found the review process to be constructive and are pleased with how the manuscript was handled editorially. We look forward to future exciting submissions from your lab.

Sincerely,
